# Labeling proteins inside living cells using external fluorophores for microscopy

Kai Wen Teng[1,2], Yuji Ishitsuka[2,3], Pin Ren[1,2], Yeoan Youn[1,2], Xiang Deng[4], Pinghua Ge[3], Sang Hak Lee[2,3], Andrew S Belmont[1,4], Paul R Selvin[1,2,3,4]*

[1]Center for Biophysics and Quantitative Biology, University of Illinois at Urbana-Champaign, Urbana, United States; [2]Center for Physics of Living Cell, University of Illinois at Urbana-Champaign, Urbana, United States; [3]Department of Physics, University of Illinois at Urbana-Champaign, Urbana, United States; [4]Department of Cell and Developmental Biology, University of Illinois at Urbana-Champaign, Urbana, United States

**Abstract** Site-specific fluorescent labeling of proteins inside live mammalian cells has been achieved by employing Streptolysin O, a bacterial toxin which forms temporary pores in the membrane and allows delivery of virtually any fluorescent probes, ranging from labeled IgG's to small ligands, with high efficiency (>85% of cells). The whole process, including recovery, takes 30 min, and the cell is ready to be imaged immediately. A variety of cell viability tests were performed after treatment with SLO to ensure that the cells have intact membranes, are able to divide, respond normally to signaling molecules, and maintains healthy organelle morphology. When combined with Oxyrase, a cell-friendly photostabilizer, a ~20x improvement in fluorescence photostability is achieved. By adding in glutathione, fluorophores are made to blink, enabling super-resolution fluorescence with 20–30 nm resolution over a long time (~30 min) under continuous illumination. Example applications in conventional and super-resolution imaging of native and transfected cells include p65 signal transduction activation, single molecule tracking of kinesin, and specific labeling of a series of nuclear and cytoplasmic protein complexes.

*For correspondence: selvin@illinois.edu

Competing interests: The authors declare that no competing interests exist.

## Introduction

Fluorescence microscopy of the living cells is often achieved through specific labeling of proteins by antibody, nanobody, or bio-specific ligand conjugated to a fluorophore. However, most of these bio-molecules and fluorophores are not able to cross the cell membrane of a living cell, making it challenging to image intracellular proteins. There are a few exceptions to this barrier, namely, by using few selected cell-permeant fluorophores that have been attached to membrane-permeant entities (*Lukinavičius et al., 2014*; *Grimm et al., 2015*; *Wombacher et al., 2010*). These have obvious trade-offs of limited choice in emission wavelengths and available class of ligands that they can be covalently attached to and maintain their selectivity for the particular intracellular protein. There has also been mixed success in delivering fluorescent probes by appending a membrane-permeant small peptide such as the TAT-TAR HIV peptide (*Silhol et al., 2002*; *Richard et al., 2002*). Another method to overcome the permeability issue is by transfecting the cells with plasmid DNA encoding the intracellular protein of interest appended to a fluorescent protein (FP). Nonetheless, there are many cases where such transfection is not possible or not desirable. In addition, the detection of FPs is less than optimal because of their limited photostability.

Other methods that attempt to overcome this limitation include microinjection, electroporation and osmotic pinosomelysis (*Okada and Rechsteiner, 1982*; *Crawford et al., 2013*; *Zhang et al., 1990*; *Kim et al., 2008*). However, these techniques have significant drawbacks such as low

throughput and the requirement of additional apparatus. Finally, two recently developed methods, such as 'biophotonic laser-assisted surgery tool (BLAST)' and 'cell squeezing' are promising techniques, although they require the cells to be cultured in specific platforms like fabricated surface or microfluidic channels (*Wu et al., 2015*; *Kollmannsperger et al., 2016*).

In this study, we make use of the well-known pore-forming bacterial toxin streptolysin O (SLO) to label intracellular proteins in mammalian cells for fluorescence microscopy applications. In the past, SLO has been used for delivering fluorescently labeled proteins—but not targeted to specific proteins (*Walev et al., 2001*). SLO has also been used with transfected cells to deliver a small lanthanide probe (*Rajapakse et al., 2010*). It has also been used for labeling RNA with 'molecular beacons' or a streptavidin-RNA probe, and also to introduce (non-fluorescent) ligands through the membrane (*Nitin and Bao, 2008*; *Kano et al., 2000*; *Santangelo et al., 2009*). Here, we present a general method for labeling intracellular proteins in transfected or non-transfected cells, in the nucleus or the cytoplasm, with probes ranging in size from small (~2 kDa) molecules to large proteins (up to 150 kDa), for either general or super-resolved fluorescence microscopy.

## Results and discussion

### Reversible permeabilization using pore-forming toxin for delivering fluorescent probes

We applied SLO to permeabilized the cells, creating pores of ~30 nm in size (*Stewart et al., 2015*) in order to deliver fluorescent probes for labeling intracellular proteins in a site-specific manner (*Figure 1a*). First, we permeabilized the cells with SLO at 37°C for 7–10 min depending on the cell confluency, followed by incubation with the fluorescent probes (i.e., fluorophores by themselves, or covalently attached to a ligand or a protein) for 5 min on ice. Excess probes were then removed by a washing step; then a 'complete medium', supplemented with ATP, GTP, and glucose, was added to reseal the membrane (see Materials and methods for specific details). The cells were then allowed to recover in the incubator at 37°C for 15–20 min. For super-resolution microscopy, namely direct-STORM (dSTORM), 4 mM glutathione and 0.5 U/mL of Oxyrase were added to create blinking fluorophores (*Heilemann et al., 2008*).

To demonstrate specific labeling of intracellular structures, we delivered three different membrane impermeant fluorescent probes (DAPI, < 1 kDa; Phalloidin-Alexa647, 1.95 kDa; Anti-PMP70-ATTO488, ~150 kDa) simultaneously into live CHO-K1 cells that have been permeabilized with SLO (*Figure 1b*). Cells that are permeabilized by SLO and labeled by the fluorescent probes should show nucleus stained by DAPI, actin filament stained by phalloidin-Alexa647, and peroxisomes stained by anti-PMP70-ATTO488. Fluorescence images of SLO treated cells (*Figure 1b*) revealed 94% (15 out of 16 cells) of the cells in the field of view contains nuclear staining by DAPI, 88% of the cells contains clear and punctate structure of labeled peroxisomes, and 88% of the cells showed filamentous structure of the actin – a stark contrast to SLO untreated negative control cells (*Figure 1b*). In addition, field of views other than the one shown in *Figure 1* were counted, and the result shows similar degree of permeabilization with a standard deviation of 8% (*N = 49 cells*). To confirm the membrane integrity after recovery, SLO-treated cells were incubated for 5 min with Texas Red dextran, (~20 kDa), a normally membrane-impermeant fluorescent molecule. The membrane of recovered cells efficiently excluded Texas Red dextran (*Figure 1b*). In contrast, the negative control cells that were not treated with SLO, showed no fluorescence signal until cells were permeabilized by SLO (*Figure 1b*). The health of the cells after permeabilization is a universal problem in any methods attempting to compromise the cell membrane. In this study, we performed several cell viability assays to demonstrate that the cells remain viable after recovering from SLO permeabilization (*Figure 1—figure supplement 1*). We showed that close to 95% of cells had intact membrane after recovery when recovered in proper buffers (*Figure 1—figure supplement 1a*); that the cells were able to divide (*Figure 1—figure supplement 1b*); and the cellular organelle displayed comparable morphology before and after SLO treatment (*Figure 1—figure supplement 1c*).

Aside from the cell viability tests performed above, we performed a translocation assay on the transcription factor p65 fused with GFP on the N-terminus, in human cancer cell line (HeLa). This would demonstrate that signal transduction events still occurred at a normal level after SLO treatment and fluorophore labeling (*Los et al., 2008*). The cells were permeabilized and intracellular p65

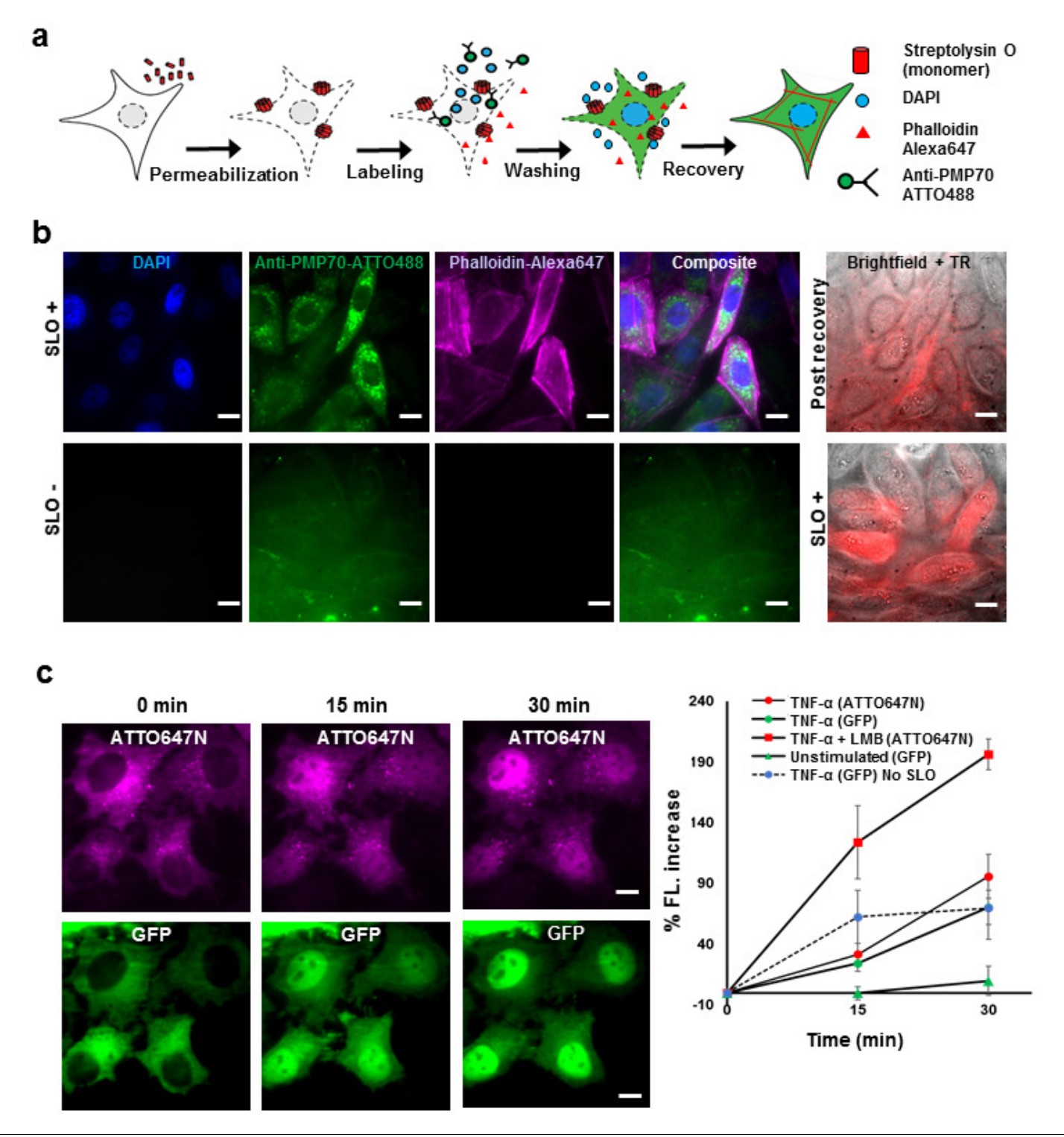

**Figure 1.** Fluorescent labeling of proteins inside the living cell with high efficiency. (a) Schematic of permeabilization, delivery, for reversible permeabilization using streptolysin O (SLO) on live CHO-K1 cells. Cells are exposed to SLO for 7–10 min before incubating with the three different fluorescent probes (DAPI, phalloidin-Alexa647, and anti-PMP70-ATTO488) on ice for 5 min. Recovery was initiated by adding 10% FBS DMEM supplemented with nucleotide ATP, GTP, and glucose. (b) Specific labeling of cellular structures by SLO delivered probes. Nucleus, peroxisome, and actin filaments are labeled by DAPI, Anti-PMP70-ATTO488, and phalloidin-Alexa647 respectively. The cells are then incubated with Texas red- 20 kDa dextran to check whether the membrane has recovered. The negative control (SLO-) shows no labeling of cellular structure. The negative control cells are then permeabilized by SLO to show the fluorescence of Texas Red in permeabilized cells. (DAPI is impermeant to CHO-K1 cells.) (c) p65

*Figure 1 continued on next page*

*Figure 1 continued*

translocating from cytosol to nucleus after SLO permeabilization in HeLa cells. TNF-α and Leptomycin B (LMB) were used to stimulate nucleus import of the p65-GFP protein labeled by GBP-ATTO647N. The nucleus fluorescence of both GFP and ATTO647N channels increased 15 and 30 min after stimulation as shown in time series fluorescence images (Left). The percent increase of nucleus fluorescence was quantified (Right). (Red square = ATTO647N intensity in the nuclear after TNF-α and LMB treatment, Green circle = GFP intensity after TNF-α treatment [*N* = 4], Red circle =ATTO647N Intensity after TNF-α treatment [*N* = 4]. Blue circle with dashed line is the GFP intensity after TNF-α treatment for cells that have never been permeabilized by SLO [*N* = 4]. Green triangle = GFP intensity without TNF-α and LMB treatments [*N* = 3]). Scale bar denotes 10 μm.

The following figure supplement is available for figure 1:

**Figure supplement 1.** Propidium iodide exclusion and cell division assay.

was labeled with an anti-GFP single-domain antibody (known as GBP [*Ries et al., 2012*]) (*Figure 1c*). When the protein TNF-α is added to the media, p65 migrates into the nucleus; if the nuclear export inhibitor-Leptomycin B (LMB) is also added, the effect is enhanced (*Los et al., 2008*). This is exactly what was quantitatively observed (*Figure 1c*). This p65-translocation result shows that the level of response to TNF-α in SLO-treated cells is comparable to untreated cells. Hence, the technique may be applied to perform similar assay on primary culture or cell lines for high-throughput drug screening.

To test the possibility of using SLO for delivering fluorescent probes ranging from <1 kDa to ~150 kDa (*Figure 2a*), we transfected the cells with plasmid encoding Halotag-GFP-ActA, a protein that binds to the outer mitochondrial membrane. Consequently, Halotag and GFP are present facing outside of the mitochondrial membrane and are accessible to our fluorescent probes. Fluorescence image of the GFP on the mitochondrial membrane (mitochondrial-GFP) is shown in *Figure 2b*. The Halotag was labeled with a cell-impermeant dye conjugated to chloroalkane, or with a biotinylated chloroalkane bound to fluorescently labeled streptavidin. Alternatively, the GFP was labeled with a GBP (a 14 kDa single domain nanobody specific to GFP), or dye-labeled-antibody (IgG). With SLO permeabilization, all of the cell-impermeant probes tested—chloroalkane-Alexa660 (<1 kDa), GBP-ATTO647N (~14 kDa), chloroalkane-biotin-streptavidin-ATTO647N (~56 kDa), and Anti-GFP IgG-labels (~150 kDa)—which varied in size from <1 KDa to ~150 kDa—were able to specifically label the mitochondrial membrane (*Figure 2*).

We then tested the accessibility of the probes to nuclear protein using the SLO delivery method. Here, we used the Proliferating Cell Nuclear Antigen (PCNA), bound to GFP (*Figure 2*). We found that GBP-ATTO647N (~14 kDa) was able to passively diffuse across the nuclear membrane, or diffuse through the nucleus pore complex, in order to label the nuclear proteins PCNA-GFP. In a separate test we found that 40 kDa FITC-dextran was able to access nucleus as well. (The fluorescence of FITC-dextran is homogenously distributed across the nucleus and cytoplasm.) In contrast, probes larger than GBP, such as streptavidin (labeled SA, 53 kDa) or 70 kDa FITC-dextran (*Figure 2—figure supplement 1a*), were not able to passively diffuse across the nuclear membrane. This cut-off for passive diffusion between 40–56 kDa is consistent with the known size-limit for diffusion into nuclear pores (*Grünwald et al., 2011*).

Surprisingly, we discovered that using small probes, <2 kDa, resulted in background-free labeling—that is, the unbound probe diffused out during the recovery phase. On the other hand, 'large' molecules, >20 kDa dextran or other similar-sized proteins, remained in the cytoplasm, creating evenly distributed fluorescence across the cytosol (*Figure 2—figure supplement 1b*). Apparently, the pore formed by SLO allows the fluorescently labeled proteins free access to entering the cell, while allowing free access for exit only for relatively small (<2 kDa) proteins. To explain this observation, we hypothesize that some of the larger probes do get washed out, but not fast enough before the membrane starts to heal, and only smaller probes are given enough time to be diffuse out completely. This background-free staining makes permeabilization by SLO advantageous over other methods like microinjection or microporation, where free probes cannot be washed away (*Hennig et al., 2015*). In addition, the required concentration of fluorescent chloroalkane with SLO-treated cells was a factor of ten lower than that recommended by the manufacturer. This might be because the manufacturer's means of getting the ligand into the cell depends on setting up a concentration gradient across the cell membrane, whereas none is necessary with SLO.

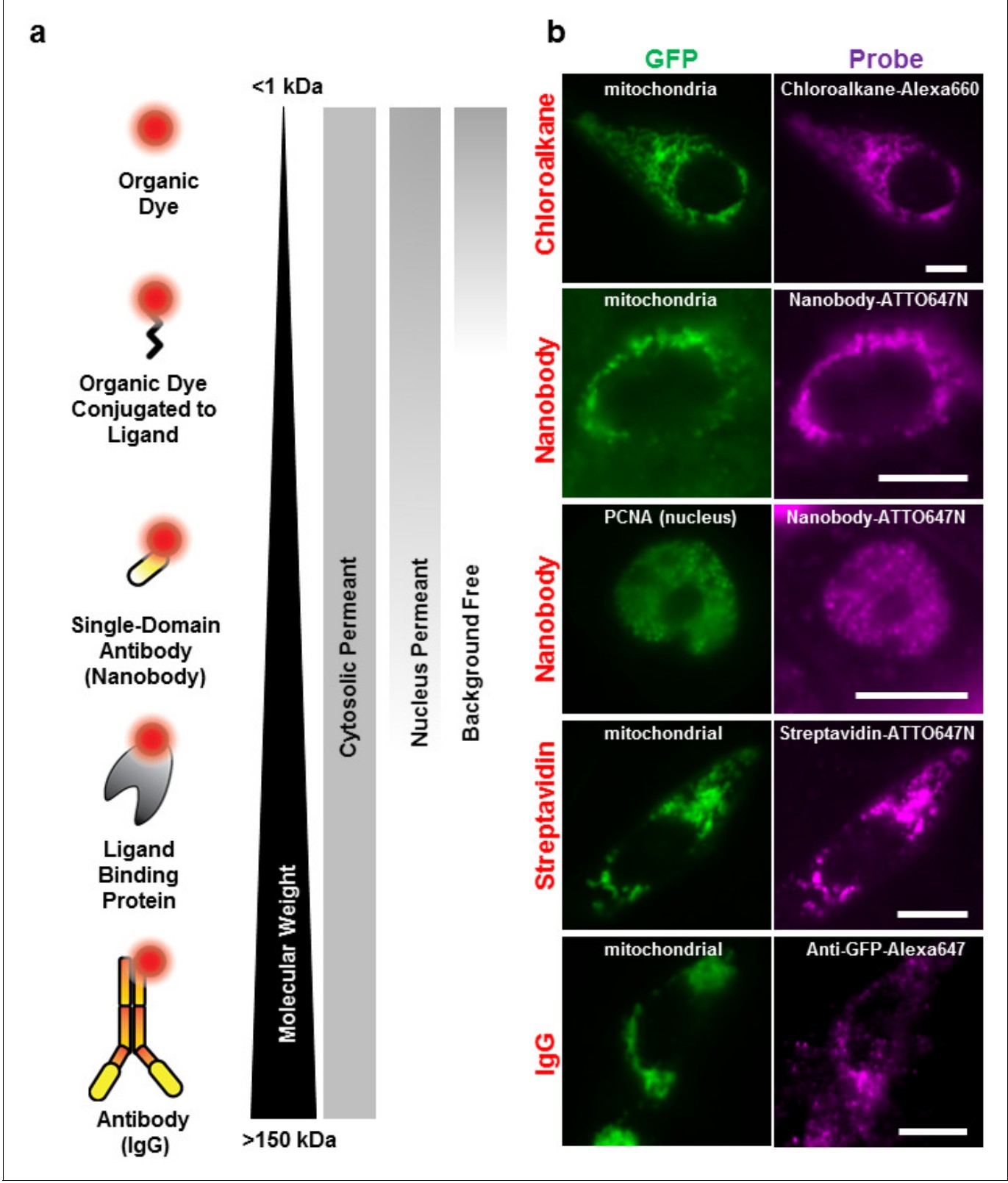

**Figure 2.** Labeling cellular proteins with a variety of probes. (a) Delivery of different sized fluorescent probes for specific labeling of intracellular proteins. The drawings depicting the fluorescent probes are not to scale. (b) Cells are labeled by various SLO delivered probes (magenta) to compare with their intracellular or nucleus target, which is tagged with GFP (green). Chloroalkane labeling was performed on U2OS cells, and the rest were done

*Figure 2 continued on next page*

*Figure 2 continued*
on CHO-K1 cells. All the probes loaded, up to the ~150 kDa IgG, were successfully delivered by SLO and labeled target proteins. However, only small proteins such as GBP (14 kDa) were able to access the nuclear target. Scale bar denotes 10 µm.
The following figure supplement is available for figure 2:

**Figure supplement 1.** The effect of fluorescent probe sizes on retention and nucleus permeability.

## Application of pore-forming toxin delivery in Super-Resolution imaging

We also demonstrate the ability to observe live cell dynamics with dSTORM (*Heilemann et al., 2008*) by looking at live U2OS human cancer cells with cell-impermeant Alexa647-Phalloidin. To cause photo-blinking necessary for dSTORM, the oxygen concentration must be kept low and a thiol-containing reducing agent must be present. 4 mM glutathione and the commercially available Oxyrase (an oxygen scavenger compatible with live cell imaging) were therefore added (*Figure 3—figure supplement 1a–b*). We measured actin dynamics for 20,000 frames at 100 ms time resolution, for a total time of 33 min (*Figure 3b*). Such tremendous photostability was the result of Oxyrase and glutathione; without them, fewer activations of Alexa647 were observed in response to a 405 nm laser. Finally, we observed a thick actin bundle split over the course of ~3 min into two separate strands, ~850 nm apart using this live super-resolution approach (*Figure 3c*, *Video 1*). Note that phalloidin is an actin stabilizer and certainly will interfere with the normal dynamic of actin. Here, we are using it as a proof-of-principle example to showcase the possibility of performing live-cell dSTORM using our technique. In addition to actin, we have also done super-resolution dSTORM imaging on Histone 2B and mitochondria labeled with Alexa647 and Alexa660 chloroalkane (*Figure 3—figure supplement 2*).

Next, we applied SLO permeabilization to enhance the signal-to-noise ratio (SNR) of the FP for in vivo single-particle tracking applications and to visualize individual fluorescent proteins amongst a large background of over-expressed FPs. FPs works very well for visualizing ensemble of proteins in cell, and constructs of FPs attached to numerous intracellular protein of interest have already been made. However, it remains difficult to study these proteins down to a single molecule level from the fluorescence signal of individual FPs due to low brightness and photostability. The goal here is to label FPs with nanobodies conjugated to dyes with photostability superior to the FPs. For instance, we show that the signal can be improved ~6-fold by detecting the signal of GBP-ATTO647N than the GFP itself (*Figure 3—figure supplement 3a–b*). Using this method, we can perform single molecule studies on a vast library of existing protein constructs with FP attached by using just FP binding nanobodies conjugated to a fluorophore with better photostability and brightness. Note that attaching multiple dyes per probe can potentially enhance the signal further; here, most of the nanobodies had one organic dye. Using this method, we tracked single kinesin molecules in the cytoplasm of U2OS cells (*Figure 3d*). We transfected the cells with 2x-mCherry-kinesin (K560), a truncated version of dimeric kinesin-1 with two mCherry's per kinesin monomer (*Norris et al., 2015*). Based on the fluorescence of mCherry alone, it is difficult to track a single kinesin protein in vivo, due to both the poor SNR of mCherry and the high background fluorescence created by out-of-focus FPs at different z-positions in the cytoplasm. Here, we enhance the kinesin signal by delivering RFP binding nanobody (RBP-ATTO647N), which binds specifically to mCherry (*Figure 3—figure supplement 3c*). Labeled individual kinesins can now easily be detected; an example trace is shown in *Figure 3e*, and an example cell showing moving kinesins labeled with ATTO647N-RBP is shown *Video 2*. The average run length and velocity was determined to be 621 ± 99 nm (Decay constant ± SE) and 1058 ± 22 nm/s (Center ± SEM) at 30°C (*Figure 3f*). Both the run length and velocity agrees with the in vitro data on single kinesin at 30°C (*Kawaguchi and Ishiwata, 2000*). In addition, we have performed the experiment at room temperature, and indeed we observed a decrease in kinesin velocity (*Figure 3—figure supplement 3d–e*). Thus, using this technique we were able to track otherwise difficult to observe single molecules in vivo, and allowed us to study the biophysical properties of molecular motors in the cytoplasm of living cells.

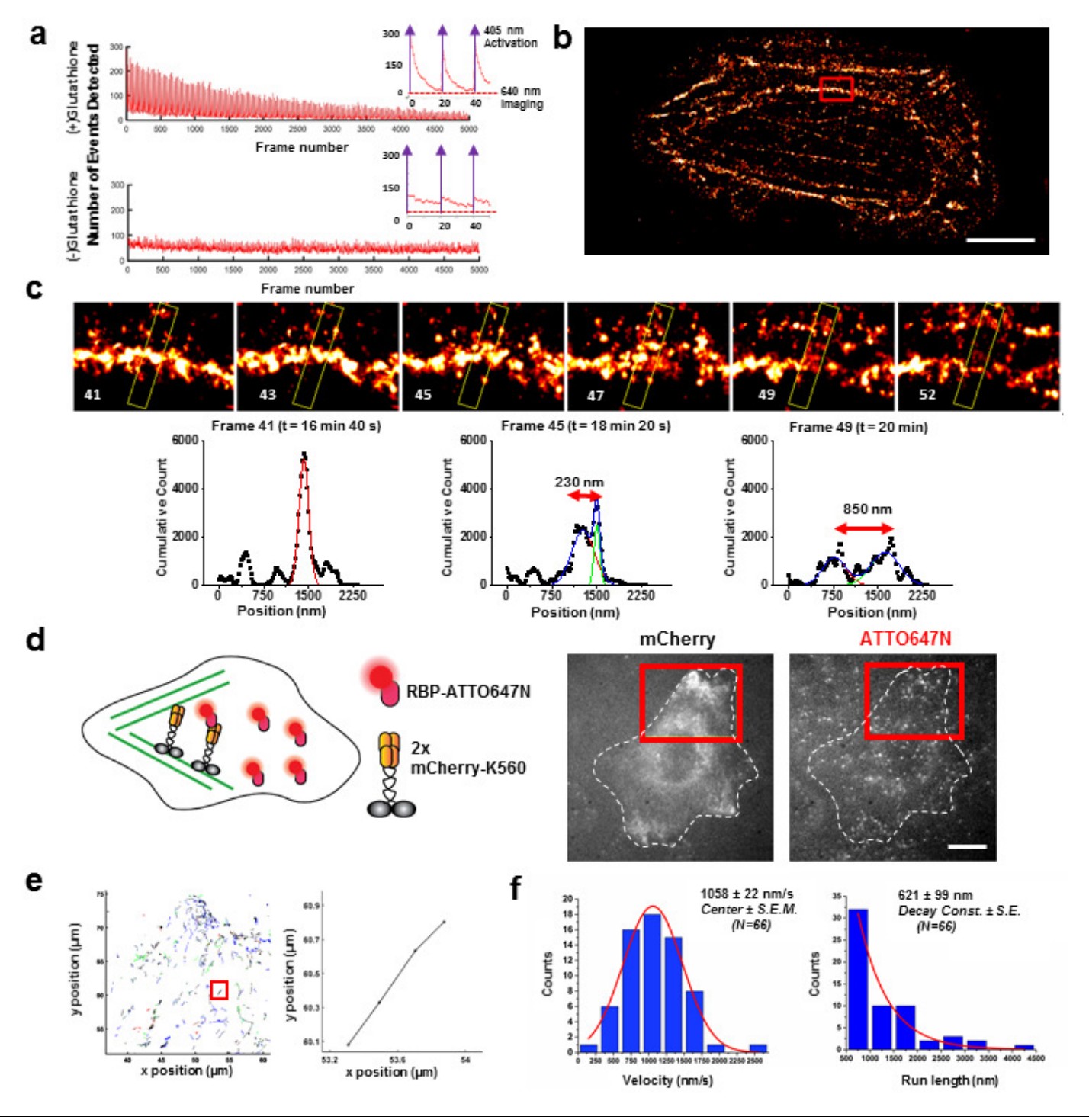

**Figure 3.** Applications of live cell protein labeling using SLO permeabilization. (**a**) The number of detected single molecule activation events per frame with and without the addition of glutathione in the recovery media. With the addition of 4 mM glutathione in the recovery media after SLO permeabilization, there are a greater number of activation events detected per frame, as well as a fewer number of detected events during illumination, suggesting that the added glutathione assists in turning off the fluorophore more efficiently. The insets show the first 50 frames of the images, where with each pulses of 405 nm laser new waves of single molecules are activated. (**b**) A dynamic dSTORM movie was acquired to examine large-scale actin dynamics. Each frame of the movie is composed of 500 frames, with a 250 frames moving average time window. The movie was recorded for a total of 20,000 frames at 100 ms exposure time, for a total acquisition time of over 30 min. Scale bar denotes 5 µm (**c**) A series of zoomed in images focusing on a small section of actins in (**b**) (yellow rectangle). A thick actin bundle can be seen to split into two thinner actin filaments. The actin filaments grew 230

*Figure 3 continued on next page*

*Figure 3 continued*

nm apart after 1 min 40 s, and then 850 nm apart after another 1 min 40 s (*Video 1*). (**d**) U2OS cells expressing 2x-mCherry-kinesin (K560) and labeled by RBP-ATTO647N. After just 3 hr of expression, the cell is filled with over-expressed mCherry-kinesin, and it is *not* possible to track individual protein. Single kinesin molecules can easily be tracked by sparsely labeling mCherry-kinesin with Red fluorescent protein Binding Protein (RBP) conjugated to ATTO647N. (**e**) Tracking individual ATTO647N-RBP labeled kinesin. Majority of kinesin trajectory in the selected area can be seen to travel uni-directionally in a short distance (<5 µm). The trajectory enclosed in the red rectangle is an example trace of detected kinesin trajectory. (**f**) To quantify the kinesin behavior in vivo, velocity and run length of labeled kinesin were measured. The kinesins are moving at an average velocity of 1058 ± 22 nm/s (Center ± SEM), and the average run length is 621 ± 99 nm (Decay constant ± SE). The measurements were taken on a heated stage at 30°C.

The following figure supplements are available for figure 3:

**Figure supplement 1.** Activation of Alexa647 in fixed and live cells in the presence and absence of Oxyrase.

**Figure supplement 2.** Additional examples of intracellular structures of living cells imaged by dSTORM using cell impermeant chloroalkane-dye that are delivered by using SLO.

**Figure supplement 3.** Labeling fluorescent protein with conjugated nanobody for kinesin tracking.

## Conclusion

We report the application of SLO permeabilization for high-throughput labeling of intracellular proteins, and the visualization of their dynamics in various biological systems. Importantly, we have applied SLO to label cells without transfection. This is achieved by labeling intracellular proteins with antibodies, nanobodies, and bio-specific ligands. This technique has tremendous application for avoiding artifacts caused by over-expression due to transfection. Furthermore, we create an environment for generating photoswitchable and photostable fluorophores in vivo for dSTORM, and enhance the signal of FPs for single particle tracking. Our SLO delivery technique can easily be combined with other signal enhancement techniques, such as 'spaghetti monster' (*Viswanathan et al., 2015*), for delivering otherwise membrane impermeant probes into living cells for labeling a single protein with numerous probes at once. Lastly, this technique is not limited to delivering of fluorescent probes; the technique can potentially be extended to other types of microscopy that relies on probe delivery, such as cryo-electron microscopy.

## Material and methods

### Cell culture

CHO-K1 cells were grown in F12K (Fisher-Mediatech) supplemented with 10% heat inactivated fetal bovine serum (HI FBS; Life Technologies) and 100 U/mL of penicillin plus 10 µg/mL streptomycin (PS; Corning). The cells were subcultured at 90–100% confluency by washing it 3x with PBS, incubate the cells with trypsin-EDTA (Corning) for 5 min, and dilute the trypsin four-fold with cell culture media.

To prepare the samples for imaging, the main cell culture was subcultured, and cells were plated in a 35 mm glass bottomed dish with 14 mm micro-well cover glass (Cellvis). The culturing of U2OS and HeLa cells followed the aforementioned protocol, except that the base media was replaced with DMEM and MEM (Fisher-Mediatech). CHO-K1, HeLa, and U2OS cells were obtained from ATCC. After the first passage, the cells were aliquoted and frozen. Each aliquot was used for no more than 30 passages. The cells were cultured under aseptic techniques and periodic examination under bright field with a 20x or 100x objective. No testing of mycoplasma contamination was performed on these cells.

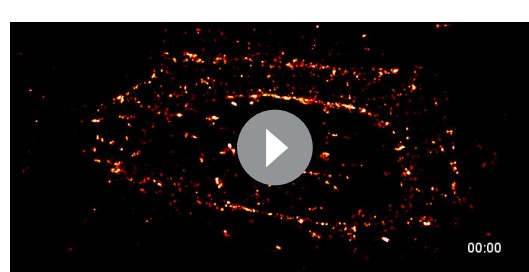

**Video 1.** Dynamic direct super-resolution imaging of actin labeled with Alexa647-phalloidin related to *Figure 3*. Real time is displayed at the bottom right hand corner of the movie in min:s.

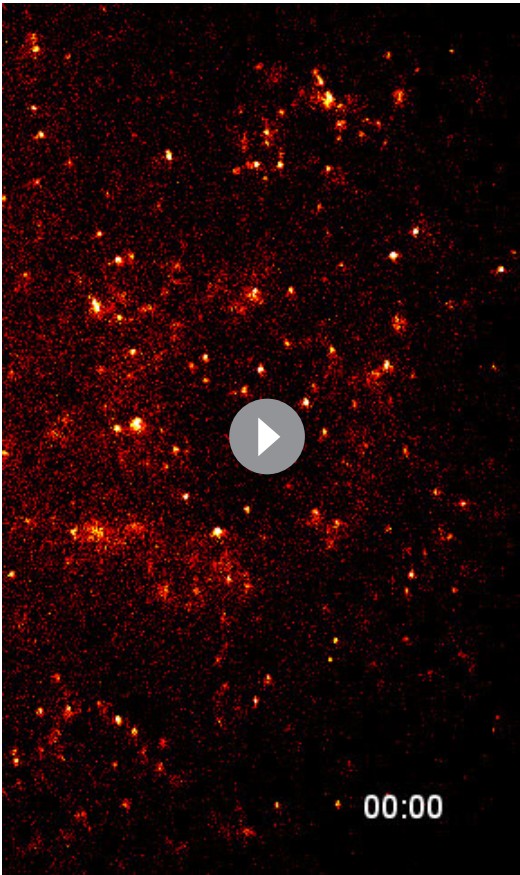

00:00

**Video 2.** Single molecule detection of individual mCherry kinesin using ATTO647N labeled anti-RFP nanobody. Real time is displayed at the bottom right hand corner of the movie in min:sec, the movie is played back in 3x the speed compared to real time.

## Reversible permeabilization and probe loading

Streptolysin O (SLO, 25,000–50,000 U, Sigma-Aldrich) was dissolved in 1 mL of molecular biology grade water. The stock solution was aliquoted and stored at −80°C. A new aliquot was thawed prior to each experiment and used within 5 hr. An aliquot of SLO stock was reduced by 10 mM TCEP (Pierce) for 20 min at 37°C. The SLO stock solution was further diluted to working concentration in DPBS supplemented with 1 mM MgCl$_2$. Different batches of SLO may have different activity, and therefore the optimal SLO concentration should first be determined. The optimized working SLO concentration was determined by testing a range of SLO concentrations for each cell line to achieve >75% permeability to 10 kDa FITC-dextran, while keeping dead cells at a minimum (<10%). Different cell lines required different dosage of SLO; a table of SLO dilution to use for a few selected cell lines and confluency is included in *Table 1*.

The cells, grown to 80–100% confluency, were incubated in SLO containing buffer for 7–10 min at 37°C. Permeabilized cells were observed under phase contrast microscopy at 20x magnification after 5 min. Granules inside the cells become more distinctive in permeabilized cells. Fluorescent probes were diluted in Tyrode's buffer without calcium (140 mM NaCl, 5 mM KCl, 1 mM MgCl$_2$, 10 mM HEPES, 10 mM glucose, pH 7.4). After SLO permeabilization, the cells were washed 2x with DPBS + 1 mM MgCl$_2$. The cells were fluorescently labeled by incubating with the probes, such as DAPI, chloroalkane-Alexa660 (Promega), Alexa647-Phalloidin (RRID:AB_2620155, Thermo Scientific), nanobody-dye conjugate (RRID:AB_2629215, Chromotek), and labeled anti-PMP70-ATTO488 IgG (RRID:AB_1841112, P0090, Sigma-Aldrich), or labeled polyclonal anti-GFP Alexa647 IgG (RRID:AB_162553, A-31852, Thermo Scientific) at a final concentration of 300–700 nM for 5 min on ice. In addition, the nanobody used in this work was either custom purified (*Wang et al., 2014*) or purchased, and could be used interchangeably. The custom nanobody was labeled through the amine group and NHS-dye. Free dyes were purified

**Table 1.** Dilution and incubation time of SLO to use for cell lines used in this study at various confluency.

| Cell line | Confluency | Dilution of SLO stock | Incubation time (min) |
|-----------|-----------|----------------------|----------------------|
| CHO-K1 | 90–100% | 250x | 7 |
| CHO-K1 | 75% | 334x | 7 |
| U2OS | 80% | 500x | 7 |
| U2OS | 50% | 774x | 7 |
| U2OS | 30% | 1000x | 7 |
| HeLa | 90% | 250x | 10 |

using 6 kDa Micro Bio-Spin 6 (Bio-Rad) or 7 kDa Zeba Spin desalting column (Thermo Scientific). For preparation of chloroalkane-SA probe, SA was incubated with biotin-PEG-chloroalkane at 1:6 molar ratio for 10 min. Free SA binding sites were then saturated with excess free biotin for 5 min and the sample was purified by using an Amicon concentrator (30 kDa MW cutoff, EMD Millipore). If the probe is prone to cause non-specific binding on the coverslip or cell membrane, 40x dilution of 10X Casein (Vector Lab) was added to the Tyrode's buffer along with the probes. After labeling, the cells were washed 3x with Tyrode's buffer, and incubated with recovery medium (Phenol red (P/R) free DMEM with sodium pyruvate, 10% FBS, 2 mM ATP, 2 mM GTP, 2 mg/mL glucose, without antibiotic) for 20 min. The idea to replace lost nucleotide triphosphate and glucose was adapted from previous publication (*Kano et al., 2012*). To check whether the cell membrane has recovered, the cells were washed with PBS 3x, and incubated with 500 nM of 20 kDa Texas Red dextran diluted in DPBS with 1 mM $MgCl_2$ for 5 min. As a negative control, another coverslip of cell underwent the same treatment except the addition of SLO was also imaged. The negative control was then exposed to SLO and consequently 20 kDa Texas Red dextran to show that the cells are permeant to 20 kDa Texas Red dextran if the cell membrane was still permeable.

## Cell viability assays

Several measures to ensure cell viability were performed on CHO-K1 cells after SLO permeabilization and recovery. The cells were recovered in different buffers such as recoveryM (10% FBS DMEM 2 mM ATP, 2 mM GTP, 2 mg/mL Glucose), 10% FBS DMEM, DMEM, Tyrode's buffer + 10% FBS, TyrodeCA (Tyrode's buffer + 2 mM $CaCl_2$), Tyrode's buffer, and DPBS. The idea of supplementing the recovery medium with different component was previously reported (*Kano et al., 2012*). For propidium iodide (PI) assay, after permeabilization with SLO cells were incubated with 1 ug/mL for 5 min on ice. At the end of staining, the cells were washed with PBS 3x. PI stained cells were imaged using 561 nm excitation and 605 nm emission (*Figure —figure supplement 1a*). A bright field image was recorded for the same region for cell counting purposes. % viability was calculated based on the number of propidium iodide stained cells over the number of cells counted in bright field.

For cell counting experiment, CHO-K1 cells were seeded on a lettered fiduciary marker coverslip (*Figure —figure supplement 1b*). The manufacturing process for the coverslip is described in literature *Lee et al., 2012*. The CHO-K1 cells were permeabilized following the protocol mentioned above with SLO. GBP-ATTO647N was delivered inside the cell without labeling any protein just to mark the cells that are permeabilized. An initial fluorescence and bright field image were recorded right after SLO permeabilization. After 16 hr, the same regions were located with the help of lettered fiduciary marker and another bright field and fluorescence image of the region was recorded. The number of cells present in this region was counted.

To examine the morphology of the cells before and after SLO permeabilization and labeling, HeLa cells were cultured on the lettered fiduciary coverslip described in the previous paragraph. The cells were transfected with plasmid encoding GFP-ActA-Halotag overnight. The cells were imaged before SLO treatment in both GFP and Alexa647 channel, and the same area was imaged after the cells were treated with SLO and the mitochondria labeled with Alexa647-Halotag ligand (*Figure 1—figure supplement 1c*).

## Plasmid and transfection

For transfection, all cells were transiently transfected with Lipofectamine 2000 (Life Technologies) following the manufacturer's protocol, with plasmid DNA (pERB254) encoding GFP-ActA-Halotag, which is a mitochondrial membrane targeted protein. pERB254 was a gift from Michael Lampson from University of Pennsylvania (Addgene plasmid # 67762) (*Ballister et al., 2015*). For PCNA-GFP used in the study, pcDNA3_NLS_eGFP_hPCNA was derived from pENeGFPPCNA (*Leonhardt et al., 2000*); eGFP-PCNA was cut from pENeGFPPCNA with BamHI and XbaI and cloned into pcDNA3 cut with HindIII and XbaI using an adapter (AGCTTATGGCTTCGTGGGGATC) to reconstruct the correct start codon for eGFP-PCNA. Unless specified otherwise, the permeabilization procedure was done on cells after overnight transfection. 2xmCherry-K560 plasmid was a gift from Verhey Lab (U. Michigan).

## Fluorescence microscopy

Fluorescence and super-resolution microscopy are performed on a Nikon Ti Eclipse microscope with 100X objective (CFI Apo TIRF 100x Oil, N.A. 1.49, Nikon). Excitation lasers (MLC400B, Agilent Technologies, 405 nm, 488 nm, 561 nm and 640 nm) were used for exciting different fluorescent probes. The images were recorded in an EMCCD (iXon DU-897E, Andor). A quad-band dichroic (ZT405-488-561-640RPC, Chroma) was used for reflectance of the excitation laser and transmission of fluorescence. The emission filter choices were 525/50 (DAPI, GFP, ATTO488), 600/50 (mCherry, Texas Red, PI), or 700/75 (Alexa647, ATTO647N). The live cell samples were imaged inside a 30°C temperature controlled chamber (InVivo Scientific) unless otherwise noted. The static conventional fluorescence images shown were averages of 10 frames taken at 50–100 ms, except for the kinesin experiment, where the spots were rapidly moving so a single frame image was shown. The cells were imaged in 10% FBS DMEM P/R free medium on a glass-bottom dish.

## p65 translocation assay

For p65 translocation assay, HeLa cell was transiently transfected with p65-GFP. 48 hr post-transfection, the cell was permeabilized with SLO, and the cells were labeled with and GBP-ATTO647N. After the cell recovered from SLO permeabilization, the cells were serum—starved in DMEM for 4 hr prior to TNF-$\alpha$ (10 ng/mL) and Leptomycin B (LMB, 2 $\mu$M) treatment.

## dSTORM with cell impermeant probe

U2OS cells were permeabilized using the protocol mentioned above. 330 nM of phalloidin-Alexa647 were loaded after permeabilization. The cells were recovered in recovery media plus 4 mM final concentration of glutathione. After the recovery from SLO permeabilization, 7.5 mL of complete media without phenol red was filled to the top of the glass bottom dish to reduce the contact of media with air. ~100 $\mu$L of Oxyrase stock solution (Oxyrase, Inc.) and 20 mM final concentration of sodium DL-lactate were added to the cell media. A series of images were recorded after the addition of Oxyrase up to a total of 2 hr. The activation laser was 405 nm (0.5 to 7.5 $\mu$W power post-objective), and Alexa647 was de-activated by shining 640 nm laser (600 $\mu$W power post-objective). For dynamic dSTORM movie, 20,000 frames were recorded per cell at 100 ms exposure time per frame. The single molecule detection analysis was done using Matlab-based algorithm, alivePALM (*Li et al., 2013*).

## Kinesin labeling and tracking in live cell

U2OS cells were transiently transfected with plasmid DNA encoding 2xmCherry K560 for 2.5–4 hr. Post-transfection the cells were permeabilized with SLO as described above. The cells were then labeled with 150x dilution of stock red fluorescent protein binding nanobody (RBP-ATTO647N, Chromotek). For each cell, a 300 frames movie was recorded using 200 ms exposure time. Spots that move processively for $\geq$3 frames without running into another molecule were cropped from the movie. For localization, the point spread function was fit to a 2D Gaussian distribution, and the position is given by the center. Localization accuracy is typically ~20 nm. The kinesin positions are rotated to linearize the displacement in one dimension. The velocity and run length is determined from the position and time of the first to last frame. In majority of our traces, the kinesin travels for <10 frames without pauses.

## Acknowledgements

This work was in part supported by NSF PHY-1430124, and NIH GM108578 to PRS, and NIH GM58460 to ASB.

## Additional information

### Funding

| Funder | Grant reference number | Author |
| --- | --- | --- |
| National Science Foundation | NSF PHY-1430124 | Kai Wen Teng Yuji Ishitsuka Pin Ren |

| | | Yeoan Youn<br>Pinghua Ge<br>Paul R Selvin<br>Sang Hak Lee |
|---|---|---|
| National Institutes of Health | NIH GM108578 | Kai Wen Teng<br>Yuji Ishitsuka<br>Pin Ren<br>Yeoan Youn<br>Pinghua Ge<br>Paul R Selvin<br>Sang Hak Lee |
| National Institutes of Health | NIH GM58460 | Xiang Deng<br>Andrew S Belmont |

The funders had no role in study design, data collection and interpretation, or the decision to submit the work for publication.

## Author contributions

KWT, Conception and design, Acquisition of data, Analysis and interpretation of data, Drafting or revising the article; YI, Conception and design, Analysis and interpretation of data, Drafting or revising the article; PR, YY, Acquisition of data, Analysis and interpretation of data; XD, Acquisition of data, Contributed unpublished essential data or reagents; PG, ASB, Drafting or revising the article, Contributed unpublished essential data or reagents; SHL, Conception and design; PRS, PI of the lab, Devised the experiments and wrote the manuscript, Conception and design, Drafting or revising the article

## Author ORCIDs

Paul R Selvin, http://orcid.org/0000-0002-3658-4218

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
