## [Decision Letter]

Thank you for submitting your article "Labeling Proteins Inside Living Cells Using External Fluorophores for Fluorescence Microscopy" for consideration by *eLife*. Your article has been reviewed by two peer reviewers, and the evaluation has been overseen by Richard Aldrich as the Reviewing Editor and the Senior Editor. The reviewers have opted to remain anonymous.

The reviewers have discussed the reviews with one another and the Reviewing Editor has drafted this decision to help you prepare a revised submission.

Summary:

In this paper Teng at al. develop a simple reversible mammalian cell permeabilization assay to deliver various fluorescent reagents to living cells. By forming small pores with the toxin Streptolysin O the authors are able to introduce probes from the extracellular solution. The authors go to great lengths to show that the labeling is efficient and does not perturb the basic biology of the cells (cell division and organelle health). In general this method will be very useful for the microscopy and biophysics community. It could become a standard labeling method. There are many applications where an easy delivery method would be useful. These include microscopy of living cells, structural mapping of proteins with FRET, drug delivery, etc.

This paper is quite straightforward and well done and we really only have several relatively small criticisms and suggestions to improve the manuscript.

Essential revisions:

1) Do some of the data (e.g., viability results) depend on the batch or source of the SLO? Or does this need to be determined anew each time SLO was ordered? I was particularly interested in this since I noticed this afternoon some suppliers sell SLO by weight rather than by units (and don't provide the activity of their SLO).

2) Can the authors comment about whether or not this method allows uptake of small quantum dots? The authors previously synthesized small quantum dots attached to nano bodies (see Wang et al., 2014) so it seems like this would be easy to test. A lot of people would be interested in ways to get Q-Dots inside live cells.

3) We really don't like the last two sentences of the Conclusion since they refer to possible applications that aren't discussed in the present manuscript so the data can't be evaluated. Those sentences should be removed and replaced with something much less specific.

4) The figure legends for the supplemental data need to be improved. Splitting each figure into individual, lettered panels would help.

5) For Figure 3—figure supplement 3. We are not sure that we agree that the authors enhanced the signal of a fluorescent protein using their method. The fluorescence of GFP was not improved, rather a protein harboring a completely different fluorophore was targeted to GFP, and it is expected that Atto647N would be much brighter than GFP. This is no different than targeting a fluorescently labeled antibody to any other protein.

6) The authors state that one of the concerns about existing methods such as microinjection or electroporation is "concern over the cell health". Surely this is also a concern with any application of SLO as well? While the authors show data that their cells are healthy after every SLO treatment, I would think that this needs to be double-checked with every new cell type tested using this method.

7) The images are quite small and hard to see. Because the main point of the paper is the quality and efficiency of labeling we would suggest showing larger images and larger fields of view. This is particularly important as the level of background labeling is a key issue in evaluating this work.

8) Reference of Kollmannsperger et al. Nat. Comm 2015 seems relevant and missing. This new work should be compared to this "cell squeezing" labeling methods.

---

## [Author Response]

*Essential revisions:*

*1) Do some of the data (e.g., viability results) depend on the batch or source of the SLO? Or does this need to be determined anew each time SLO was ordered? I was particularly interested in this since I noticed this afternoon some suppliers sell SLO by weight rather than by units (and don't provide the activity of their SLO).*

The activity of SLO will need to be determined each time a new SLO is ordered from the manufacturer. However, we have found that if the SLO comes from the same lot number, the activity of the SLO is consistent. We have tried using SLO that’s supplied by weight, and converted from weight to activity unit using manufacturer’s conversion factor, but we have found that the activity converted from weight is different (in our case by factor of 2) compared to SLO supplied by activity. Therefore, we suggest in the Methods section that the labs who wish to use this technique should test the activity of SLO by titrating the concentration of SLO needed to induce >75% of permeabilization to a marker molecule (e.g. 10 kD dextran), while keeping >90% of the cells alive (as tested by PI staining) after the recovery step. We do agree with the reviewers that we should caution readers about the batch to batch variation. We added a statement of caution in our method, and the new sentences now read:

“Different batches of SLO may have different activity, and therefore the optimal SLO concentration should first be determined. The optimized working SLO concentration was determined by testing a range of SLO concentrations for each cell line to achieve >75% permeability to 10 kDa FITC-dextran, while keeping dead cells at a minimum (<10%).”

*2) Can the authors comment about whether or not this method allows uptake of small quantum dots? The authors previously synthesized small quantum dots attached to nano bodies (see Wang et al., 2014) so it seems like this would be easy to test. A lot of people would be interested in ways to get Q-Dots inside live cells.*

We are pleased that the reviewers recognize the expertise of our lab. The possibility of using small quantum dot to efficiently label intracellular proteins was in fact one of the driving motivation behind this project. At this point, we had some preliminary luck getting small quantum dots inside, but it turned out that it requires a different type of coating to bring them in efficiently. The standard coating used in reference 25 requires blocking agents such as casein or BSA to prevent it from sticking to the cell membrane non-specifically, and by doing so, we suspect that it increases the size of the quantum dots. We now have a new type of quantum dot with a different polymer coating that does not require the use of blocking proteins. With these, we could deliver small quantum dots inside cells in an amount that is clearly not caused by endocytosis. However, we are still working on functionalizing chemistry of these small quantum dots for specific protein targeting. Since the emphasis of the paper is to label intracellular proteins and because we have not proven that intracellular proteins were labeled with the small quantum dots, we decided not to include it in this manuscript.

*3) We really don't like the last two sentences of the Conclusion since they refer to possible applications that aren't discussed in the present manuscript so the data can't be evaluated. Those sentences should be removed and replaced with something much less specific.*

We agree with the reviewers that the last sentence in the conclusion was not well phrased. We have rewritten the sentence with a more reserved tone for suggesting a potential new application. The newly revised sentence now reads as below:

“Lastly, this technique is not limited to delivering of fluorescent probes; the technique can potentially be extended to other types of microscopy that relies on probe delivery, such as cryo-electron microscopy.”

*4) The figure legends for the supplemental data need to be improved. Splitting each figure into individual, lettered panels would help.*

We thank the reviewers for pointing this out and suggesting an improvement to the readability of our paper. We have slightly enlarged supplemental figure 1. We have divided our supplemental figures into lettered panels, particularly for supplemental figure 2 and 3. Figure legends have been modified to address each figure panels.

*5) For Figure 3—figure supplement 3. We are not sure that we agree that the authors enhanced the signal of a fluorescent protein using their method. The fluorescence of GFP was not improved, rather a protein harboring a completely different fluorophore was targeted to GFP, and it is expected that Atto647N would be much brighter than GFP. This is no different than targeting a fluorescently labeled antibody to any other protein.*

We agree with the reviewers that the signal of the GFP is not enhanced. A more accurate description is that we went from observing GFP to observing a more photostable dye that is bound to the GFP. We added a short excerpt of narrative in our main text clarifying the motivation of this application, and it reads:

“FPs works very well for visualizing ensemble of proteins in cell, and constructs of FPs attached to numerous intracellular protein of interest have already been made. However, it remains difficult to study these proteins down to a single molecule level from the fluorescence signal of individual FPs due to low brightness and photostability. The goal here is to label FPs with nanobodies conjugated to dyes with photostability superior to the FPs. For instance, we show that the signal can be improved ~6-fold by detecting the signal of GBP-ATTO647N than the GFP itself (Figure 3—figure supplement 3). Using this method, we can perform single molecule studies on a vast library of existing protein constructs with FP attached by using just FP binding nanobodies conjugated to a fluorophore with better photostability and brightness.”

We’ve also made small modification to the figure legend of Figure 3—figure supplement 3, and it now reads:

“Improving the detection of proteins attached to fluorescent protein by using nanobody labeling”.

*6) The authors state that one of the concerns about existing methods such as microinjection or electroporation is "concern over the cell health". Surely this is also a concern with any application of SLO as well? While the authors show data that their cells are healthy after every SLO treatment, I would think that this needs to be double-checked with every new cell type tested using this method.*

We agree with the reviewers that the integrity of the membrane and cell health is not an exclusive problem to microinjection and electroporation. We’ve added a new sentence in our Results section, and it reads:

“The health of the cells after permeabilization is a universal problem in any methods attempting to temporarily compromise the cell membrane. To test the cell health, we performed several cell viability assays to demonstrate that the cells remain viable after recovering from SLO permeabilization (Figure 1—figure supplement 1).” We’ve also removed the statement that “concern over the cell health” as an exclusive problem of microinjection and electroporation.

*7) The images are quite small and hard to see. Because the main point of the paper is the quality and efficiency of labeling we would suggest showing larger images and larger fields of view. This is particularly important as the level of background labeling is a key issue in evaluating this work.*

We agree with the reviewers that the fluorescence images in the paper can be further improved by increasing the size of the images. We have made size modifications throughout the text. For Figure 1, the fluorescence image of the cell has been increased by roughly 1.4x the original size. For Figure 2, the size of the fluorescence images has been increased by 1.6x. To make the images fit in one figure, we have changed the arrangement of the figure from horizontal to vertical arrangement. The size of the images has been increased for several figures in the supplemental figures as well. As for larger field of view, most of the images in the paper, besides Figure 2, have images that are already shown in the full field of view taken by 100x or 150x magnification. Figure 2 is shown in region of interest to show the features of and quality of mitochondria labeling. In addition, we also have addressed the issue noted below, which stated that green/magenta display may be more friendly to the viewers who are red/green color blind. We have replaced all the figures with green/red to green/magenta color scheme.

*8) Reference of Kollmannsperger et al. Nat. Comm 2015 seems relevant and missing. This new work should be compared to this "cell squeezing" labeling methods.*

We thank the reviewers for pointing out a recently developed technique that we have overlooked. We have added the reference to the technique alongside another recently developed technique “BLAST”. The new paragraph now reads:

“Finally, two recently developed methods, biophotonic laser-assisted surgery tool (BLAST) and cell squeezing, are promising techniques, although they require the cells to be cultured in specific platforms like fabricated surface or microfluidic channels.”